

# Non-stationary Extreme Value Analysis: a simplified approach for Earth science applications

Lorenzo Mentaschi[1,2], Michalis Vousdoukas[1], Evangelos Voukouvalas[1], Ludovica Sartini[2], Luc Feyen[1], Giovanni Besio[2], Lorenzo Alfieri[1]

[1]European Commission, Joint Research Centre (JRC), Institute for Environment and Sustainability (IES), Climate Risk Management Unit, via Enrico Fermi 2749, 21027 Ispra, Italy
[2]Università di Genova, Dipartimento di Ingegneria Chimica, Civile ed Ambientale, via Montallegro 1, 16145 Genova, Italy

*Correspondence to*: Lorenzo Mentaschi (lorenzo.mentaschi@jrc.ec.europa.eu)

**Abstract.** Statistical approaches to study extreme events require by definition long time series of data. The climate is subject to natural and anthropogenic variations at different temporal scales, leaving their footprint on the frequency and intensity of climatic and hydrological extremes, therefore assumption of stationarity is violated and alternative methods to conventional stationary Extreme Value Analysis (EVA) need to be adopted. In this study we introduce the Transformed-Stationary (TS) methodology for non-stationary EVA. This approach consists in (i) transforming a non-stationary time series into a stationary one to which the stationary EVA theory can be applied; and (ii) reverse-transforming the result into a non-stationary extreme value distribution. As a transformation we propose and discuss a simple time-varying normalization of the signal and show that it allows a comprehensive formulation of non stationary GEV/GPD models with constant shape parameter. A validation of the methodology is carried out on time series of significant wave height, residual water level, and river discharge, which show varying degrees of long-term and seasonal variability. The results from the proposed approach are comparable with the ones from (a) a stationary EVA on quasi-stationary slices of non stationary series and (b) the previously applied non stationary EVA approach. However, the proposed technique comes with advantages in both cases, as in contrast to (a) it uses the whole time horizon of the series for the estimation of the extremes, allowing for a more accurate estimation of large return levels; and with respect to (b) it decouples the detection of non-stationary patterns from the fitting of the extreme values distribution. As a result the steps of the analysis are simplified and intermediate diagnostics are possible. In particular the transformation can be carried out by means of simple statistical techniques such as low-pass filters based on running mean and standard deviation, and the fitting procedure is a stationary one with a few degrees of freedom and easy to implement and control. An open-source MATLAB toolbox has been developed to cover this methodology, available at https://bitbucket.org/menta78/tseva.

## 1 Introduction

Extreme Values Analysis (EVA) attains a great importance in several applied sciences, particularly in Earth Science, because it is a fundamental tool to study the magnitude and frequency of extreme events, and changes therein (e.g. Alfieri et al., 2015; Forzieri et al., 2014; Jongman et al., 2014; Resio and Irish, 2015; Vousdoukas et al., 2016). Climatic extreme events



are usually associated to disasters and damages with relevant social and economic cost. A correct statistical evaluation of the strength of extreme events related to their average return period is crucial for impact assessment, for the evaluation of the risks affecting human lives and activities, and for planning actions connected to risk management and prevention (Jongman et al., 2014).

Often it is required to apply EVA to non-stationary time series, i.e. series with statistical properties varying in time due to changes in the dynamic system. In particular, relevant climate changes are usually associated to variations in the statistical properties of time series of climatic variables. For example an intensification of the meridional thermal gradient at middle latitudes on global scale would lead to an increase of the climatic variability (e.g. Brierley and Fedorov, 2010) which would involve a reduction of the average return period of storms with a given strength. Consequently in the study of climate

changes an accurate statistical estimation of middle-long term extremes is inherently connected to the application of non-stationary methodologies.

While a general theory about non stationary EVA has not yet been formulated (Coles, 2001) there are several studies describing methodologies for the estimation of time-varying extreme value distributions on non stationary time series, which rely on the pragmatic approach of using the standard extreme value theory as a basic model that can be enhanced by means

of further statistical techniques (e.g. Coles, 2001; Davison and Smith, 1990; Husler, 1984; Leadbetter, 1983; Méndez et al., 2006).

An established technique consists in expressing the parameters of an extreme value distribution as time-varying parametric functions $M$ of time, for some custom parameters ($\alpha_i$, $\beta_i$, $\gamma_i$ …). By means of a fitting process such as the Maximum Likelihood Estimator (MLE) it is then possible to fit the values of ($\alpha_i$, $\beta_i$, $\gamma_i$ …) to model the extremes of the non-stationary

series. Appropriate implementations of such a methodology, hereinafter referred to as "established approach" and abbreviated as EA, produce meaningful results, as proved by a number of contributions (e.g. Cheng et al., 2014; Gilleland and Katz, 2015; Izaguirre et al., 2011; Méndez et al., 2006; Menéndez et al., 2009; Mudersbach and Jensen, 2010; Russo et al., 2014; Sartini et al., 2015; Serafin and Ruggiero, 2014).

A drawback of this approach is that there is no general indication on how to formulate the function $M$. As a rule the model

should be parsimonious, i.e. simpler models should be preferred. For this reason often several test formulations of $M$ are used together, and then the best model is chosen through a balance between high likelihood and low degrees of freedom, for example by means of the Akaike criterion (Akaike, 1973). Furthermore the choice of $M$ depends on the statistical model we choose for the extreme value analysis: for example for the same series the $M$ used for the Generalized Extreme Value (GEV) model is different from the $M$ used for the Generalized Pareto Distribution (GPD) model. As in this approach the estimation

of the time-varying properties of the series is incorporated into the fitting of the extreme value distribution, non-stationary fitting methods are required despite being relatively complex to implement and control.

Another widespread approach to deal with non-stationary series is dividing them into quasi-stationary slices and applying the stationary theory to each slice (e.g. Vousdoukas et al., 2016). This technique will be hereinafter referred to as "stationary on slice" and abbreviated as SS. Although this technique allows to detect meaningful trends for short return periods, its use has





the drawback of reducing the size of the sample used for the EVA, implying larger uncertainty in the estimation of long return periods.

This research aims to contribute to the field of non-stationary EVA by introducing the Transformed-Stationary extreme value methodology, hereinafter referred to as TS, which allows to decouple the study of the non stationary behavior of the series

from the fit of the extreme value distribution. To this purpose it introduces a standard methodology to model the variations of the statistical properties of the series.

In section 2.1 the TS methodology is discussed and outlined in a general and theoretic way, while section 2.2 describes its implementation. Section 3 is dedicated to the validation of the methodology, and section 4 illustrates a comparison with other widespread approaches for the EVA of non stationary series such as EA and SS for modeling time series characterized

by seasonal cycles and time series showing long term trends. In section 5 the results are discussed and in section 6 some conclusions are drawn.

## 2    Methods and data

### 2.1    Theoretical background

The TS methodology consists in three steps: transforming of a non-stationary time series $y(t)$ into a stationary series $x(t)$,

performing a stationary Extreme Value Analysis (EVA), and back-transforming the resulting extreme value distribution into a time dependent one.

The transformation $y(t) \rightarrow x(t)$ we propose is:

$$x(t) = f(y,t) = \frac{y(t) - tr_y(t)}{ca_y(t)} \ . \tag{1}$$

where $tr_y(t)$ is the trend of the series, i.e. a curve representing the long term, slowly varying tendency of the series, and $ca_y(t)$ is the long term, slowly varying amplitude of a confidence interval which represents the amplitude of the distribution

of y(t). In particular, if $ca_y(t)$ is equal to the long term varying standard deviation $std_y(t)$ of the series y(t), Eq. (1) reduces to a simple time-varying renormalization of the signal:

$$x(t) = f(y,t) = \frac{y(t) - tr_y(t)}{std_y(t)} \ . \tag{2}$$

In the rest of the manuscript for simplicity we will limit our analysis to Eq. (2), knowing that all the considerations can be easily extended to any time varying confidence interval $ca_y(t)$.

Transformation (2) guarantees that the average of $x(t)$ and its standard deviation are uniform in time, which is a necessary

condition for $x(t)$ to be stationary. In particular the transformed signal $x(t)$ has null average and variance equal to 1. It is





worth noting that the transformed series $x(t)$ is not necessarily stationary: a series with a constant trend and a uniform standard deviation may still have a time-dependent auto-covariance which would invalidate the hypothesis of stationarity. Before proceeding with the analysis, a stationarity test should be carried out to ensure that $x(t)$ is stationary and that its annual maxima can be fitted by a stationary extreme value distribution. A simple test can be performed for example ensuring

that higher order statistics such as skewness and kurtosis are roughly constant along the series.

Once the hypothesis of stationarity of $x(t)$ is verified we can estimate the GEV $G_X(x)$ best fitting its extremes, for example through a Maximum Likelihood Estimator (MLE). $G_X(x)$ is then given by

$$G_X(x) = \Pr(X < x) = \exp\left\{-\left[1 + \varepsilon_x\left(\frac{x - \mu_x}{\sigma_x}\right)\right]^{-1/\varepsilon_x}\right\} \qquad (3)$$

where the shape, scale and location parameters $\varepsilon_x$, $\sigma_x$ and $\mu_x$ do not depend on time. To find the time dependent distribution $G_Y(y,t)$ fitting the non stationary time series $y(t)$ we note that:

$$G_Y(y) = \Pr[Y(t) < y] = \Pr[f^{-1}(X,t) < y] = \Pr[X < f(y,t)] = G_X[f(y,t)], \qquad (4)$$

where $f(y,t)$ is the transformation from y to x given by Eq. (1), and $f^{-1}(x,t)$ is its inverse,

$$f^{-1}(x,t) = y(t) = std_y(t) \cdot x + tr_y(t), \qquad (5)$$

It is always possible to compute $G_Y(y,t)$ from $G_X(x)$ because $f(y,t)$ is a monotonically increasing function of $y$ for every time $t$, being the standard deviation $std_y(t)$ always positive.

Using Eqs. (3) and (5) in Eq. (4) we find

$$G_Y(y,t) = G_X[f(y,t)] = \exp\left\{-\left[1 + \varepsilon_x\left(\frac{\dfrac{y - tr_y(t)}{std_y(t)} - \mu_x}{\sigma_x}\right)\right]^{-1/\varepsilon_x}\right\} =$$

$$= \exp\left\{-\left[1 + \varepsilon_x\left(\frac{y - tr_y(t) - \mu_x \cdot std_y(t)}{\sigma_x \cdot std_y(t)}\right)^{-1/\varepsilon_x}\right]\right\}. \qquad (6)$$

Therefore if $x(t)$ is fitted by the stationary GEV $G_X(x)$ then $y(t)$ is fitted by the time dependent GEV $G_Y(y,t)$ with

shape, scale and location parameters given by





$$\varepsilon_y = \varepsilon_x \,,$$
$$\sigma_y(t) = std_y(t) \cdot \sigma_x \,,$$
$$\mu_y(t) = std_y(t) \cdot \mu_x + tr_y(t) \tag{7}$$

It can be shown that the time-dependent GEV parameters given by Eq. (7) are the same that would be obtained from a non-stationary MLE on the series $y(t)$ in order to fit the parametric expressions of $\varepsilon_{ns}$, $\sigma_{ns}$ and $\mu_{ns}$ given by

$$\varepsilon_{ns} = const. \,,$$
$$\sigma_{ns} = std_y(t) \cdot a \,,$$
$$\mu_{ns} = std_y(t) \cdot b + tr_y(t) \tag{8}$$

for varying parameters $a$ and $b$. In fact if $p_{GX}(x)$ is the pdf associated to the distribution $G_X(x)$, then the MLE for $G_X(x)$ is estimated so that

$$\sum \log[\, p_{GX}(x)] = \max \,, \tag{9}$$

which involves, considering for example the scale parameter $\sigma_x$

$$\sum \frac{\partial}{\partial \sigma_x} \log[\, p_{GX}(x, \sigma_x)] = 0 \,. \tag{10}$$

In the non-stationary MLE what is maximized is the log-likelihood of the non stationary pdf $p_{Gns}(y,t)$ varying the parameters $a$ and $b$. For example considering the parameter $a$ we impose

$$\sum \frac{\partial}{\partial a} \log[\, p_{Gns}(y, a, t)] = 0 \tag{11}$$

Let us assume that $p_{Gns}(y,t)$ coincides with the pdf $p_{GY}(y,t)$ associated to the GEV $G_Y(y,t)$ given by (6) and that $a = \sigma_x$. Considering that

$$p_{GY}(y,t) = \frac{\partial}{\partial y} G_Y(y,t) = p_{GX}(x) \frac{\partial}{\partial y} f(y,t) = \frac{p_{GX}(x)}{std_y(t)} \tag{12}$$

we obtain

$$\sum \frac{\partial}{\partial a} \log[\, p_{Gns}(y, a, t)] = \sum \frac{\partial}{\partial \sigma_x} \log[\, p_{GY}(y, \sigma_x, t)] = \sum \frac{\partial}{\partial \sigma_x} \log \left[ \frac{p_{GX}(x, \sigma_x)}{std_y(t)} \right] =$$
$$\sum \frac{\partial}{\partial \sigma_x} \left\{ \log[\, p_{GX}(x, \sigma_x)] - \log[std_y(t)] \right\} = \sum \frac{\partial}{\partial \sigma_x} \log[\, p_{GX}(x, \sigma_x)] = 0 \tag{13}$$

where the last step is possible because $std_y(t)$ does not depend on $\sigma_x$.




The same principle can be applied differentiating $\sum \log[p_{GY}(x,\mu x,t)]=0$ on the location parameter $\mu_x$ to maximize the log-likelihood, finding the condition

$$\sum \frac{\partial}{\partial \mu_x}\log[p_{GY}(x,\mu_x,t)] = \sum \frac{\partial}{\partial \mu_x}\log[p_{GX}(x,\mu_x)] = 0 \tag{14}$$

This means that if x is stationary, when the likelihood is maximum for $p_{GX}(x)$ it is maximum also for $p_{GY}(y,t)$, and that applying an MLE to best fit the stationary parameters $(\sigma_x,\mu_x)$ coincides to applying a non-stationary MLE to best fit the parameters (a, b) of the parametric expression (8). The equivalence between the two methodologies suggests that the TS approach is dual to EA, meaning that any implementation of EA is equivalent to an implementation of the TS approach for some transformation $f(y,t): y(t) \rightarrow x(t)$ (see appendix A for a more detailed discussion). One can also prove that Eq. (1) allows a general TS formulation with constant shape parameter, i.e. all the TS models with a constant $\varepsilon_y$ can be connected to Eq. (1) (see appendix A). This last result is remarkable, because it shows that Eq. (1) is exhaustive for all the TS models with constant shape parameter.

The findings drawn above are general and can be applied also to Peak Over Threshold (POT) methodologies, because the GPD is formally derived from the GEV as the conditional probability that an observation beyond a given threshold $u$ is greater than x. In particular, the POT/GPD parameters are given by

$$u_y(t) = std_y(t) \cdot u_x + tr_y(t),$$
$$\varepsilon_y = \varepsilon_x = const.,$$
$$\sigma_{GPDy}(t) = \sigma_y(t) + \varepsilon_y[u_y(t) - \mu_y(t)] = std_y(t) \cdot \sigma_{GPDx} \tag{15}$$

where $u_x(t)$ and $u_y(t)$ are the thresholds of the x and y time series, $\varepsilon_y = \varepsilon_x$ is the shape parameter, $\sigma_{GPDx}$ and $\sigma_{GPDy}(t)$ are the GPD scale parameters of x and y, $\sigma_y$ and $\mu_y$ are the scale and location parameters of a GEV associated to the GPD, and have been included in Eq. (15) to make it clear how the parameter $\sigma_{GPDy}(t)$ can be derived.

It worth noting that the TS methodology is "neutral" for a stationary series, i.e., the application of this methodology to a stationary series leads to the same results as a stationary EVA with the same underlying statistical model. That is because in such case $tr_y$ and $std_y$ are constant, and transformation (2) reduces to a constant translation and scaling.

## 2.1.1 Modelling the seasonality

In general we would like to model the fact that extreme events vary with season, with a typical size of local winter extremes different from that of local summer. A simple way to add the seasonal cycle to formulation (7-15) is expressing the trend $tr_y(t)$ and the standard deviation $sn_{std}(t)$ as




$$tr_y(t) = tr_{0y}(t) + sn_{tr}(t) \, ,$$

$$std_y(t) = std_{0y}(t) \cdot sn_{std}(t) \tag{16}$$

where $tr_{0y}(t)$ and $sn_{tr}(t)$ are respectively the slowly varying and seasonal components of the trend, $std_y(t)$ is the long term varying standard deviation and $sn_{std}(t)$ is the seasonality factor of the standard deviation. Applying Eq. (16) to (2) we obtain

$$x(t) = \frac{y(t) - tr_{0y}(t) - sn_{tr}(t)}{std_{0y}(t) \cdot sn_{std}(t)} \, . \tag{17}$$

The time varying GEV parameters can be expressed as

$$\varepsilon_y = \varepsilon_x = const. \, ,$$

$$\sigma_y(t) = std_{0y}(t) \cdot sn_{std}(t) \cdot \sigma_x \, , \tag{18}$$

$$\mu_y(t) = std_{0y}(t) \cdot sn_{std}(t) \cdot \mu_x + tr_{0y}(t) + sn_{tr}(t)$$

and the time varying POT/GPD parameters can be expressed as

$$u_y(t) = std_{0y}(t) \cdot sn_{std}(t) \cdot u_x + tr_{0y}(t) + sn_{tr}(t) \, ,$$

$$\varepsilon_y = \varepsilon_x = const. \, , \tag{19}$$

$$\sigma_{GPDy}(t) = std_{0y}(t) \cdot sn_{std}(t) \cdot \sigma_{GPDx} \, .$$

## 2.2 Implementation

The implementation of the TS methodology is illustrated in Figure 1. The fundamental input is represented by the series itself, and the core of the implementation consists of a set of algorithms for the elaboration of the time varying trend $tr_{0y}(t)$, standard deviation $std_{0y}(t)$ and seasonality terms $sn_{tr}(t)$ and $sn_{std}(t)$.

In this study we propose algorithms based on running means and running statistics (see section 2.2.1). Hence an important aspect is the definition of a time window $T$ for the estimation of the long term statistics $tr_{0y}(t)$ and $std_{0y}(t)$, and of a time window $T_{sn}$ for the estimation of the seasonality. The computation of $tr_{0y}(t)$ and $std_{0y}(t)$ acts as a low-pass filter removing the variability within $T$. Therefore $T$ should be chosen short enough to incorporate in the analysis the variability above the desired time scale but long enough to exclude noise, short term variability and sharp variations of the statistical properties of the transformed series. For example in studies about long term climate changes a reasonable choice is imposing $T$=30 years, because this is the generally accepted time horizon for observing significant variations in the climate (e.g. Arguez and Vose, 2011; Hirabayashi et al., 2013). It is worth stressing that the chosen value of $T$ should be verified a-posteriori to ensure that the transformed series is stationary. The time window $T_{sn}$ is used to estimate the intra-annual variability of the standard deviation (see section 2.2.1). In Figure 1 the input corresponding to the seasonal time window $T_{sn}$ is drawn in a dashed box





because its value is relatively easier to choose than that of $T$. For the examined case studies a value of two months for $T_{sn}$ always resulted in a satisfactory estimation of the seasonal cycle.

In this implementation of the TS methodology the estimation of the long term statistics is separated from the estimation of the seasonality. This separation allows both the study of the sole long term variability of the extreme values, which is the

usual approach studying the extremes on an annual basis, and of the combination of long term and seasonal variability, which is the usual approach studying the extremes on a monthly basis.

After the estimation of $tr_{0y}(t)$, $std_{0y}(t)$, $sn_{tr}(t)$ and $sn_{std}(t)$ we can apply Eq. (2) and perform a stationary EVA on the transformed series. It is important to stress that the stationary EVA is performed on the whole time horizon. The stationarity of the transformed signal allows us to apply different techniques for the EVA. In this study we illustrate the GEV and GPD

approaches, but an interesting development would be the elaboration of non-stationary techniques for other approaches such as (Goda, 1988) or (Boccotti, 2000) based on the TS methodology.

The final step of the implementation is the back-transformation of the fitted extreme value distribution into a non stationary one as given by Eq. (8) and (18) for GEV and by Eq. (15) and (19) for GPD.

### 2.2.1   Estimation of trend, standard deviation and seasonality

There are several possible ways of estimating the slowly varying trend and standard deviation and their seasonality. We propose here a simple methodology based on running mean and standard deviation. We formulate the trend $tr_{0y}(t)$ as a running mean of the signal $y(t)$ on a multi-yearly time window $T$,

$$tr_{oy}(t) = \sum_{tt=t-T/2}^{tt=t+T/2} y(tt) / N_t \; , \tag{20}$$

where $N_t$ is the number of observations available during the time interval $[t - T/2, t + T/2]$. The seasonality of the trend can be estimated as the monthly mean of the de-trended series. For a given month of the year the seasonality is then

$$sn_{tr}[month(t)] = \sum_{years} \frac{[y(tt) - tr_{0y}(tt)]\big|_{tt \in month(t)}}{N_{month}} \; , \tag{21}$$

where the subscript $tt \in month(t)$ indicates that the averaging operation is limited to time intervals within each considered month of the year. For example the seasonality of January is computed as the average on all the Januaries of the detrended signal. To estimate the slowly varying standard deviation we execute a running standard deviation with the same time window used to estimate $tr_{0y}(t)$:

$$std_{oy}(t)\big|_{ROUGH} = \sum_{tt=t-T/2}^{tt=t+T/2} \sqrt{[y(tt) - \bar{y}(tt \in [t - T/2, t + T/2])]^2 / N_{Tsn}} \; . \tag{22}$$





Where the subscript "rough" stresses the fact that this expression is sensitive to outliers and that its direct employment leads to a relevant statistical error, as it will be explained in session 2.2.2. To overcome this problem we smooth $std_{oy}(t)\big|_{ROUGH}$ with a moving average on a time window smaller than $T$, for example T/S with S=2:

$$std_{oy}(t) = \sum_{tt=t-T/2S}^{tt=t+T/2S} S \; std_{0y}(tt)\big|_{ROUGH} \; / \, N_t \; . \tag{23}$$

It is worth stressing that in general a further smoothing of the results of running means and standard deviations is licit if it reduces the error and improves the detection of the slowly varying statistical behavior of the time series. This is because the estimation of $tr_{0y}(t)$ and $std_{0y}(t)$ consists in a low-pass filter which result should be smooth on time scales lower than $T$ and affected by low relative error.

To estimate the seasonality we perform another running standard deviation $std_{sn}(t)$ on a time window $T_{sn}$ much shorter than one year, in the order of the month,

$$std_{sn}(t) = \sum_{tt=t-T_{sn}/2}^{tt=t+T_{sn}/2} \sqrt{[y(tt) - \overline{y}(tt \in [t - T_{sn}/2, t + T_{sn}/2])]^2 / N_t} \; . \tag{24}$$

The seasonality of the standard deviation can be then computed as the monthly average of the ratio between $std_{sn}(t)$ and $std_{0y}(t)$:

$$sn_{std}[month(t)] = \sum_{years} \frac{[std_{sn}(tt)/std_{0y}(tt)]\big|_{tt \in month(t)}}{N_{tt \in month(t)}} \; . \tag{25}$$

The estimated seasonality terms $sn_{tr}$ and $sn_{std}$ are periodic with a period of one year. In order to smooth them and remove any possible noise in the signal, we take into account only their first three Fourier components computed in a period of one year, corresponding to components with a periodicity of one year, six months and three months.

### 2.2.2 Statistical error

Since there is an inherent error in the estimation of trend, standard deviation and seasonality given by Eqs. (21-25) we need to estimate it and propagate it to the statistical error of the parameters of the non-stationary GEV and GPD distributions. In general, given a sample $s$ of data with size $N$, average $\overline{s}$, variance $var(s)$ and standard deviation $std(s)$ we have[1]:

$$var(\overline{s}) = var(s)/N \Rightarrow err(\overline{s}) = std(s)\big/\sqrt{N} \; , \tag{26}$$

---

[1] We can evaluate the error on the average by propagating to expression $\overline{s} = \sum s_i \big/ N$ the intrinsic error of each observation, given by the standard deviation of s. The error on the standard deviation can be evaluated considering that in a Gaussian approximation the quantity $S = \sum_N s_i^2 / var(s)$ follows a chi squared distribution with standard deviation 2N.





$$\mathrm{var}[\mathrm{var}(\ s)] \approx 2\,\mathrm{var}(\ x)^2/N \Rightarrow err\,[std\,(s)] \approx std\,(s) \cdot \sqrt[4]{2/N}\ . \tag{27}$$

Using Eqs. (26) and (27) we can estimate the error on $tr_{0y}(t)$ and $std_{0y}(t)\big|_{ROUGH}$ as

$$err(tr_{0y}) \approx std_{0y}\big/\sqrt{N_t}\ , \tag{28}$$

$$err[std_{0y}]\big|_{ROUGH} \approx std_{0y} \cdot \sqrt[4]{2/N_t}\ . \tag{29}$$

As mentioned in session 2.2.1 Eq. (29) tends to return rather high values of the error relative to $std_{0y}(t)$. For example if we are considering a time window of 20 years with an observation every 3 hours we have

$$N_t \approx 59000 \quad \Rightarrow \quad \frac{err[std_{0y}]\big|_{ROUGH}}{std_{0y}} \approx 7.6\%\ . \tag{30}$$

Using expression (23) for the estimation of $std_{0y}(t)$ overcomes this issue because we can estimate the uncertainty on

$std_{0y}(t)$ as the error on the standard deviation averaged on the time window $T/S$, which is significantly lower than the error given by Eq. (30). Using Eq. (26) we find

$$err[std_{0y}] \approx \frac{err[std_{07}]\big|_{ROUGH}}{\sqrt{N_t/S}} = std_{0y} \cdot \sqrt[4]{\frac{2S^2}{N_t^3}}\ . \tag{31}$$

We can estimate the error on the seasonality of the trend $sn_{tr}$ by adding the error estimated for $tr_{0y}(t)$ to the one due to the monthly mean. As the statistical error of independent Gaussian variables sums vectorially we obtain:

$$err(sn_{tr}) = \sqrt{err^2[mntmean(y)] + err^2(tr_{0y})}\ , \tag{32}$$

where the *mntmean(y)* operator represents the monthly average of y. If for example one considers the month of January, it is

the average computed on all the Januaries of the time series. Assuming the error on *mntmean(y)* as approximately constant within the year, it follows that

$$err[mntmean(y)] \approx std_{0y}\big/\sqrt{N_{month}} \approx std_{0y} \cdot \sqrt{12/N_{tot}}\ , \tag{33}$$

where $N_{month}$ is the number of observations corresponding to the considered month, $N_{tot}$ is the total number of elements of the series *y(t)*, $N_{month} \approx N_{tot}/12$. Therefore Eq. (32) can be rewritten as

$$err(sn_{tr}) \approx std_{0y}\sqrt{12/N_{tot} + 1/N_t}\ . \tag{34}$$

The error on $sn_{std}$ can be estimated as the error of the average ratio $std_{sn}/std_{0y}$. Using Eq. (27) the error of the ratio

$std_{sn}/std_{0y}$ is given by



$$err\left(\frac{std_{sn}}{std_{0y}}\right) \approx \sqrt{\left[\frac{err(std_{sn})}{std_{0y}}\right]^2 + \left[\frac{std_{sn}}{std_{0y}^2}err(std_{0y})\right]^2} \approx$$

$$\frac{std_{sn}}{std_{0y}}\sqrt{\sqrt{\frac{2}{N_{sn}}} + \sqrt{\frac{2S^2}{N_t^3}}} \approx sn_{std}\sqrt[4]{\frac{2}{N_{sn}}} \, , \qquad (35)$$

where $N_{sn}$ is the average number of observations within the time window $T_{sn}$ and assuming $N_t \gg N_{sn}$. We can then estimate the error on $sn_{std}$ as the error of the monthly average of $std_{sn}/std_{0y}$ :

$$err(sn_{std}) \approx err\left(\frac{std_{sn}}{std_{0y}}\right)\Bigg/\sqrt{N_{month}} \approx sn_{std}\sqrt{\frac{12}{N_{tot}}}\sqrt[4]{\frac{2}{N_{sn}}} = sn_{std}\sqrt[4]{\frac{288}{N_{tot}^2 N_{sn}}} \, . \qquad (36)$$

Using Eqs. (29), (34) and (36) we can estimate the error on the time varying GEV parameters as

$$err(\varepsilon_y) = err(\varepsilon_x) \, ,$$
$$err(\sigma_y) = \sqrt{[std_{0y} \cdot sn_{std} \cdot err(\sigma_x)]^2 + [std_{0y} \cdot err(sn_{std}) \cdot \sigma_x]^2 + [err(std_{0y}) \cdot sn_{std} \cdot \sigma_x]^2} \, ,$$
$$err(\mu_y) = \sqrt{[std_{0y} \cdot sn_{std} \cdot err(\mu_x)]^2 + [std_{0y} \cdot err(sn_{std}) \cdot \mu_x]^2 + [err(std_{0y}) \cdot sn_{std} \cdot \mu_x]^2 + err^2(tr_{0y}) + err^2(sn_{tr})} \, ,$$

$$(37)$$

and the error on the time varying GPD parameters as

$$err(u_y) = \sqrt{[std_{0y} \cdot sn_{std} \cdot err(u_x)]^2 + [std_{0y} \cdot err(sn_{std}) \cdot u_x]^2 + [err(std_{0y}) \cdot sn_{std} \cdot u_x]^2 + err^2(tr_{0y}) + err^2(sn_{tr})} \, ,$$
$$err(\varepsilon_y) = err(\varepsilon_x) \, ,$$
$$err(\sigma_{GPDy}) = \sqrt{[std_{0y} \cdot sn_{std} \cdot err(\sigma_{GPDx})]^2 + [std_{0y} \cdot err(sn_{std}) \cdot \sigma_{GPDx}]^2 + [err(std_{0y}) \cdot sn_{std} \cdot \sigma_{GPDx}]^2} \, .$$

$$(38)$$

## 2.3   Data and validation

To assess the generality of the approach the TS methodology has been validated on time series of different variables, from different sources and with different statistical properties.

The analysis of annual and monthly maxima has been carried out on time series of significant wave height at two locations, the first located in the Atlantic Ocean, West of Ireland (coordinates -10.533°E, 55.366°N) the second close to Cape Horn

(coordinates 60.237°E, -57.397°N). The data have been obtained by means of wave simulations performed with the spectral model Wavewatch III® (Tolman, 2014) forced by the wind data projections of the RCP8.5 scenario (van Vuuren et al., 2011) of the CMIP5 model GFDL-ESM2M (Dunne et al., 2012) on a time horizon spanning from 1970 to 2100. This dataset will be hereinafter referred to as GWWIII. Here the TS methodology is applied to examine its applicability to climate change studies.

The annual and monthly analysis have been repeated on a series of water level residuals offshore of the Hebrides Islands (Scotland, coordinates -7.9E, 57.3N) obtained from a 35 years hindcast of storm surges at European scale (Vousdoukas et al.,



2016) forced by the ERA-INTERIM reanalysis data (Dee et al., 2011). This dataset will be hereinafter referenced as
JRCSURGES.

For annual maxima we furthermore compare the TS methodology with the SS technique as, for example, implemented by
(Alfieri et al., 2015) and (Vousdoukas et al., 2016). To this purpose we extracted time series from projections of streamflow

in the Rhine and Po rivers covering a time horizon from 1970 to 2100 (Alfieri et al., 2015) hereinafter referred to as
JRCRIVER. Also the two series of significant wave height of West Ireland and Cape Horn extracted from the GWWIII
dataset have been employed in this comparison.

Finally we compare the TS methodology and the EA for monthly maxima using time series of significant wave height
extracted from a 35-years wave hindcast database (Mentaschi et al., 2015) in proximity of the locations of La Spezia and

10 Ortona. The analysis of this dataset, hereinafter referred to as WWIII_MED, focuses on a comparison between seasonal
cycles modeled by the two approaches.

## 3    Results

### 3.1    Waves: annual extremes

The validation of the TS methodology was performed first on the time series of significant wave height of West Ireland and

15 Cape Horn from the GWWIII dataset. We verified first the non seasonal transformation given by Eq. (2) and the time
dependent GEV/GPD given by Eqs. (7) and (15). By neglecting the seasonality, this formulation is suitable to find extremes
and peaks on an annual basis. For technical reasons the two series do not have data in two time intervals, from 2005 to 2010
and from 2092 to 2095, but the impact of the missing data on the analysis is small specially if we choose a time window $T$
large enough for the estimation of the trend and of the standard deviation using Eqs. (20) and (22). In particular for this

analysis we chose a time window of 20 years, which is long enough to ensure the accuracy of the results and short enough to
include the multidecadal variability of a 130 years long time series.

The results of the analysis for the two time series are illustrated respectively in Figure 2 and Figure 3. Panel (a) of each
figure shows the original time series and its slowly varying trend and standard deviation. Panel (b) illustrates the normalized
series obtained through the transformation 1, allowing an evaluation "at a glance" of the stationarity of the normalized series.

The mean and the standard deviation of the normalized series plotted in panel (b) are respectively equal to 0 and 1 due to the
normalizing procedure. Higher order statistics such as the skewness and the kurtosis are included in the graphics to support
the assumption of stationarity of the normalized series. From the normalized time series we extracted the annual maxima and
estimated the corresponding non-stationary GEV as given by Eq. (7) (see panel (c) of Figure 2 and Figure 3). Moreover we
performed a Peak Over Threshold (POT) selection of the extreme events on the normalized series by selecting the threshold

in order to have on average 5 events per year, following (Ruggiero et al., 2010), corresponding for both of the series to the
97[th] percentile. From the resultant POT sample we estimated the corresponding non-stationary GPD as given by Eq. (15) (see
panel (d) of Figure 2 and Figure 3). In panels (c) and (d) the shape parameters ε estimated by the MLE for the GEV and the



GPD are also reported. Inter-decadal oscillations in the annual maxima are modeled for both of the series, though they are more pronounced for the West Ireland time series. Moreover, for both the series there is a tendency of the annual maxima to increase, more pronounced for the series of Cape Horn, where the increase in the annual maxima of significant wave height estimated by GWWIII is of about 2 m.

It is worth noting that for both the considered series the statistical mode of GEV and GPD grows faster in time than the slowly varying trend $tr_y(t)$. This is due to the fact that the growth of the location parameter $\mu_y(t)$ of the non stationary GEV (expression 7), and of the threshold $u_y(t)$ of the non stationary GPD (Eq. 15) are related not only to the growth of $tr_y(t)$ but also to the growth of $std_y(t)$. The high tail of the distributions grows even faster because also the scale parameter is proportional to $std_y(t)$.

The impact of the statistical error of the slowly varying trend and standard deviation on the uncertainty of the distribution parameters have been examined using expressions (37) and (38), which for the non seasonal analysis reduce to

$$err(\varepsilon_y) = err(\varepsilon_x),$$
$$err(\sigma_y) = \sqrt{[std_y \cdot err(\sigma_x)]^2 + [err(std_y) \cdot \sigma_x]^2},$$
$$err(\mu_y) = \sqrt{[std_y \cdot err(\mu_x)]^2 + [err(std_y) \cdot \mu_x]^2 + err^2(tr_y)}, \tag{39}$$

for the GEV, and to

$$err(u_y) = \sqrt{[std_y \cdot err(u_x)]^2 + [err(std_y) \cdot u_x]^2 + err^2(tr_y)},$$
$$err(\varepsilon_y) = err(\varepsilon_x),$$
$$err(\sigma_{GPDy}) = \sqrt{[std_y \cdot err(\sigma_{GPDx})]^2 + [err(std_y) \cdot \sigma_{GPDx}]^2}, \tag{40}$$

for the GPD. The result is that for the non seasonal analysis the error due to the estimation of trend and standard deviation is negligible with respect to the error associated to the stationary MLE. In Table 1 the values of the different components of the error compared in Eqs. (39) and (40) are reported together with the total error estimated for each parameter of the non

stationary GEV and GPD. Since the threshold $u_x$ of the stationary GPD was selected to have on average 5 events per year, the error has been computed as the uncertainty related to this definition. The percentage contribution to the squared error is also reported in Table 1, in a single column because the percentages estimated for the two series are roughly equal. The error for both GEV and GPD and for both of the series is clearly dominated by the error associated to the estimation of the

parameters of the stationary distributions ($[std_y \cdot err(\sigma_x)]$ and $[std_y \cdot err(\mu_x)]$ for the GEV and $[std_y \cdot err(\sigma_{GPDx})]$ and $[std_y \cdot err(u_x)]$ for the GPD).





### 3.2 Waves: monthly extremes

The seasonal formulation of the approach is suitable to estimate extreme value distributions on a monthly basis. Hence, we applied Eq. (17) to estimate the normalized series, fitted a stationary GEV of monthly maxima by means of a MLE and back-transformed into a non stationary GEV through Eq. (18). It is worth stressing that for the stationary MLE the entire normalized series was used, covering a time horizon of 130 years. For the GPD we selected the threshold in order to have on average 12 events per year, corresponding to the 93[th] percentile for both of the series. Results are displayed in Figure 4 for the location of West Ireland and in Figure 5 for Cape Horn. To make the seasonal cycle distinguishable in the figures we plotted only a slice of 5 years from 2085 to 2090. The meaning of the four panels in Figure 4 and Figure 5 are the same as in Figure 2 and 3. The non stationary extreme value distribution estimated for the location of West Ireland presents a strong seasonal cycle with extremes higher and more broad-banded during winter. For Cape Horn the seasonal cycle is weaker, with extremes of significant wave height slightly lower during the local summer. The estimated seasonal GEV and GPD are significantly lower than those estimated for the non-seasonal analysis because in the seasonal analysis we consider monthly extremes, while in the non-seasonal one we consider annual extremes.

It is worth stressing that in the study of the monthly maxima the long term trend is also estimated, even if it cannot be appreciated in Figure 4 and Figure 5 due to the short time horizon represented.

Table 2 reports the components of the statistical error due to the uncertainty in the estimation of the seasonality together with the components due to the stationary MLE. The components of the error due to the uncertainty in the estimation of $tr_{0y}$ and $std_{0y}$ were omitted as they are negligible as compared with the error associated to the fitting of the stationary extreme value distribution (see section 3.1). In Table 2 we can see that, as for the non-seasonal analysis, the error for both GEV and GPD and for both series is clearly dominated by the uncertainty associated to the estimation of the parameters of the stationary distributions, though in this case the error related to the stationary MLE is significantly smaller than that found for the non-seasonal analysis, due to the larger sample of data.

### 3.3 Residual water levels

To verify the performance of the TS methodology on a series from a different source, of a different quantity and with different statistical characteristics, we tested it on a series of water level residuals extracted from the JRCSURGES dataset for a location offshore of the Hebrides Islands, Scotland, with coordinates (-7.9E, 57.3N). This series is characterized by a flat trend $tr_y(t)$ because the model results are approximately constant-averaged. Therefore almost all the variability is modeled by the TS methodology in the standard deviation $std_y(t)$. Since the time horizon of this series is shorter than that of the GWWIII projections we chose a time window for the computation of the trend of 6 years to better identify its interannual variability. The results of the TS analysis of the yearly maxima are shown in Figure 6. The series displays also a





strong seasonal behavior with annual maxima usually occurring during the local winter (for brevity the seasonal analysis is not illustrated).

An interesting aspect is that the estimated standard deviation $std_y(t)$ presents a strong correlation ($\rho=0.79$) with the annual means of the North Atlantic Oscillation (NAO) index. This is illustrated in Figure 7, where the scatter plot of $std_y(t)$ versus the annual means of the NAO index (panel a) and the two time series (panel b) are represented. As a consequence the estimated annual maxima are also correlated with the NAO index.

## 4  Comparison with other approaches

### 4.1  Stationary methodology on time slices for long trend estimation

A comparison was carried out between the TS methodology and the SS technique, which consists in performing a stationary analysis on quasi-stationary slices of data. This analysis was carried out on river discharge projections for the Po and the Rhine river extracted from the JRCRIVER dataset and on the projections of significant wave height extracted from the GWWIII dataset for the locations of West Ireland and Cape Horn. The TS methodology was applied with a time window of 30 years to estimate a non stationary GPD of annual maxima. The SS technique was carried out using a GPD approach on time slices of 30 years from 1970 to 2000, from 2020 to 2050 and from 2070 to 2100. For both of the methodologies the threshold was selected to have on average 5 peaks per year.

Results are illustrated in Figure 8, where the return levels of the projected discharge of the Rhine river are illustrated for three time slices. In the figure, the continuous black line and the green band represent the return levels and the 95% confidence interval estimated by the TS methodology, the dashed black line represents the return levels estimated by the stationary EVA on the considered slice (labeled in the legend as SS). As expected the return levels estimated for short return periods by the two methodologies are close, while they tend to spread for high return periods. This fact is also evident from Figure 9, where the return levels estimated by the two methodologies are plotted one versus the other for the river discharge of the Rhine and the Po and for the significant wave height of West Ireland and Cape Horn. We can see that the two methodologies for the analyzed time series are in good agreement for return periods below 30 while they spread for larger return levels. Some quantitative figures about this fact are reported in Table 3, where is reported the normalized bias NBI of the return levels of the two methodologies, defined as

$$NBI = mean(RL_{TS} - RL_{cmp})/mean(RL_{cmp}),\tag{41}$$

where $RL_{TS}$ and $RL_{cmp}$ are the return levels returned respectively by the TS and the SS methodologies. Table 3 also includes the maximum deviation between the return levels estimated by the TS and by the SS methodology. The NBI and the maximum deviations were obtained comparing results of the two techniques on the three 30-year time windows. From Table 3 we can see that the maximum deviation for return periods up to 30 years is always below 6%, while for higher return period it increases up to 13% for the discharge of the Po river. This is mainly due to the fact that for the stationary analysis



on the quasi-stationary time slices we consider a sample of only 30 years, which leads to large uncertainty ranges in the estimation of large return periods such as 100 and 300 years. This also explains the sharp variations of high return levels that we find between the three time windows using the SS approach. These variations are likely more related to the uncertainty in estimating the levels associated to long return periods rather than to climatic changes. The TS methodology allows a more

accurate estimation of high return levels because it uses the whole sample of 130 years, and this represents one of the strengths of using the TS methodology instead of SS.

## 4.2    Established non-stationary approach for seasonal variability

Section 3 shows that the TS methodology is mathematically equivalent to a particular implementation of the EA methodology as described for example by (Coles, 2001; Izaguirre et al., 2011; Menéndez et al., 2009; Sartini et al., 2015).

For the sake of completeness in this paragraph we show the results of a comparison between the performances of a different formulation of the EA methodology. In its formulation the parameters of the non stationary GEV of the monthly maxima are expressed as

$$\mu(t) = \beta_0 + \sum_{i=1}^{N_\mu} [\beta_{2i-1} \cos(i\omega t) + \beta_{2i-1} \sin(i\omega t)]$$

$$\sigma(t) = \alpha_0 + \sum_{i=1}^{N_\sigma} [\alpha_{2i-1} \cos(i\omega t) + \alpha_{2i-1} \sin(i\omega t)]$$

$$\varepsilon(t) = \gamma_0 + \sum_{i=1}^{N_\varepsilon} [\gamma_{2i-1} \cos(i\omega t) + \gamma_{2i-1} \sin(i\omega t)] \tag{42}$$

where $\beta_0$, $\alpha_0$ and $\gamma_0$ are the stationary components, $\beta_i$, $\alpha_i$ and $\gamma_i$ are the harmonics amplitudes, $\omega = 2\pi T^{-1}$ is the angular frequency, with $T$ corresponding to one year, $N_\mu$, $N_\sigma$ and $N_\varepsilon$ are the number of harmonics and $t$ is expressed in years. The

parameters $\beta_i$, $\alpha_i$ and $\gamma_i$ have been therefore optimized through a non-stationary MLE in order to fit the monthly maxima of the non-stationary series. Different combinations of $N_\mu$, $N_\psi$ and $N_\varepsilon$ have been tested and the best model was chosen as the one presenting the lowest value of the Akaike criterion (Akaike, 1973) given by

$$AIC = 2k - 2\log(L), \tag{43}$$

where $k$ is the number of degree of freedoms of the model, $L$ is the likelihood. In particular the maximum value tested for $N_\mu$, and $N_\psi$ is 3 while the maximum considered value of $N_\varepsilon$ is 2. In general this model can be extended to incorporate long term

trends, but the two series examined in this test display flat trends, hence Eq. (42) is adequate to model them.

In the comparison, the EA and the seasonal TS methodology (GEV only) were applied to the same series of significant wave heights relative to the WWIII_MED dataset described in section 2.3. For the transformed-stationary approach a 10-year time window was used for the computation of the long-term trend. The results of the two methodologies are similar, with a roughly flat trend and a strong seasonal pattern. The comparison of the seasonal cycles estimated by the two techniques is

represented in Figure 10 for the two series. In the figures the continuous red and green lines are the location and scale parameters (μ and σ respectively) as estimated by the TS approach. The dashed red and green lines are the location and scale




parameters estimated through the EA. The blue dots represent the monthly maxima, while the color scale represents the time varying probability density estimated by the transformed-stationary methodology. Since for both of the series the Akaike criterion selected models with a constant shape parameter $\varepsilon$, these are reported in the figure for both of the series together with those estimated by the TS methodology.

The GEV parameters estimated by the two approaches are in good agreement, and the small differences have relatively small impact on the return levels, as one can see in Figure 11 where the return levels estimated by the two methodologies for the month of January are plotted. For both of the series the return levels estimated by EA lie within the 95% confidence interval estimated by the TS methodology. Table 4 reports the values of normalized bias NBI between the return levels estimated by the TS and the EA methodologies, defined as in Eq. (41). In the table the values of $NBI$ are reported for the four seasons for

return periods of 5, 10, 30, 50 and 100 years, and for both La Spezia and Ortona. In the used definition of seasons, Winter starts on December 1[st], Spring on March 1[st], Summer on June 1[st] and Autumn on September 1[st]. We did not report return levels of periods greater than 100 years because the extension of the data covers only 35 years, and the estimates for such periods are inaccurate for both the methodologies. The average deviation between $RL_{TS}$ and $RL_{cmp}$ for the considered time series are rather small, below 7% for all of the seasons.

**5    Discussion**

Extreme Value Analysis is a subject of broad interest not only for Earth Science, but also for other disciplines such as Economy and Finance (e.g. Gençay and Selçuk, 2004; Russo et al., 2015), Sociology (e.g. Feuerverger and Hall, 1999), Geology (e.g. Caers et al. 1996), and Biology (e.g. Williams, 1995), among others. As a consequence non-stationarity of signals is a common problem (e.g. Gilleland and Ribatet, 2014). In this respect it is important to stress that the TS

methodology is general, and its applicability does not require a time series for any specific property but the stationarity of the transformed signal. Therefore even if in this study the technique was applied only to series related to Earth Science, it can be employed in all the disciplines dealing with extremes.

Given that the extreme value statistical model is an important component of applications like the ones presently discussed (e.g. Coles, 2001; Hamdi et al., 2013), it is important to stress that the theory was formulated in a way that is not restricted to

GEV and GPD, but can be extended to any other statistical model for extreme values. In particular, since the GEV distribution is a generalization of the Gumbel, Frechet and Weibull statistics, TS can be reformulated separately for these three distributions; as well as for the r-largest approach statistics which have been also commonly used (e.g. Coles, 2001; Hamdi et al., 2013). Finally an extension of TS to statistical models not based on the GEV theory (e.g. Boccotti, 2000; Goda, 1988) may open the way to their non-stationary generalization and could be an interesting direction for future research.

The presently discussed approach was presented using the trend, standard deviation and seasonality to perform a simple, time-varying normalization of the signal, allowing different types of analysis. The first product of the methodology is related to the estimation the extreme values of the signal. In addition, the TS approach allows the analysis of the long term




variability; and as an example it was shown to be useful in relating the long term trend of the signal with the NAO climatic index (see section 3.3). Finding correlations of natural parameters with climatic indices is a theme of common interest in Earth Science, especially in view of climate change (e.g. Barnard et al., 2015; Dodet et al., 2010; Plomaritis et al., 2015). If a time series is long-term correlated to a climatic index, an advantage of the TS methodology is that it is able to model

extremes correlated to the index without considering explicitly it in the computation. Finally, the TS methodology was also extended to describe the seasonal variability of the extremes which is also critical for climate studies (e.g. Sartini et al. 2015; Menendez et al. 2009; Méndez et al. 2006).

As shown in section 4 the TS methodology comes with advantages over both the SS methodology (e.g. Vousdoukas et al. 2016) and the EA (e.g. Cheng et al., 2014; Gilleland and Katz, 2015; Izaguirre et al., 2011; Méndez et al., 2006; Menéndez

et al., 2009; Mudersbach and Jensen, 2010; Russo et al., 2014; Sartini et al., 2015) in terms of accuracy of the results and of conceptual and implementation simplicity. In particular in the comparison with the SS methodology for long term variability the return levels estimated by the two techniques are similar for return periods for which the SS is accurate. The use of the whole time horizon of the series represents a major advantage of the TS over the SS methodology, because it allows more accurate estimations of the return levels associated to long return periods. A conceptual advantage of the TS methodology

over the EA is that it decouples the detection of the non-stationary behavior of the series from the best fit of the extreme value distribution: the goal of estimating the time-varying statistical features of the series is delegated to the transformation. This fact provides a simple diagnostic tool to evaluate the validity of the model applied to a particular series: the model is valid if the transformed series is stationary. This simple diagnostic is useful to validate the output of the approach. Moreover this decoupling simplifies both the detection of non stationary patterns and the fitting of the extreme values distribution. In

particular the detection of non stationary patterns can be accomplished by means of simple statistical techniques such as low-pass filters based on running mean and standard deviation, and the fit of the extreme value distribution can be obtained through a stationary MLE with a small number of degrees of freedom, easy to implement and control. Moreover, unlike many implementation of the EA (e.g. Cheng et al., 2014; Gilleland and Katz, 2015; Izaguirre et al., 2011; Méndez et al., 2006; Menéndez et al., 2009; Sartini et al., 2015; Serafin and Ruggiero, 2014) the detection of non stationary patterns

illustrated in this manuscript does not require an input parametric function M for the variability, making the TS methodology well suited for massive applications with the simultaneous evaluation of lots of time series, for which a common definition of M would be difficult.

It is worth remarking that the EA implemented for example using Eq. (42) is able to model a shape parameter varying in time, while the TS methodology using transformation (1) is not. While in principle this is a weak point of the TS

methodology described in this manuscript, assuming a constant shape parameter is most of cases a reasonable assumption, because in general simple models should be preferred to complex ones (e.g. Coles, 2001). In particular using the EA the Akaike criterion (Akaike, 1973), which favors simple models with fewer degrees of freedoms, often selects models with fixed shape parameter (e.g. Sartini et al. 2015; Menendez et al. 2009). Moreover, the finding that a non stationary GEV




always corresponds to a transformation of the non stationary time series into a stationary one, shown in appendix A, suggests that a generalization of the TS methodology is possible in order to include models with time varying shape parameters.

## 6    Conclusions

In this manuscript the TS methodology for non-stationary extreme value analysis is described. The main assumption underlying this approach is that if a non stationary time series can be transformed into a stationary one to which the stationary EVA theory can be applied, then the result can be back-transformed into a non-stationary extreme value distribution through the inverse transformation. The proposed methodology is general, and even if in this study we applied it only to series related to Earth Science, it can be employed in all the sciences dealing with EVA. Moreover, though we discussed it only for GEV and GPD, it can be extended to any other statistical model for extremes.

As a transformation we proposed a simple time-varying normalization of the signal, estimated by means of time-varying mean and standard deviation. This simple transformation was also adapted to describe the seasonal variability of the extremes. In addition it was proved to provide a comprehensive model for non stationary GEV and GPD with constant shape parameter, meaning that it can be applied to wide range of non-stationary processes. The formal duality between the TS approach and the established one has also been proved, suggesting that a complete generalization of the TS approach is possible to include models with time-varying shape parameter.

The methodology was tested on time series of different sources, sizes and statistical properties. An evaluation of the statistical error associated to the transformation led to the conclusion that for the examined series it is negligible (the squared error is 2 orders of magnitude smaller) with respect to the error associated to the stationary MLE and, for GPD, to the estimation of the threshold.

The TS methodology was compared with the technique of performing a stationary EVA on quasi-stationary slices of non stationary series (SS methodology) for the estimation of the long term variability of the extremes, and with the established approach (EA) to non stationary EVA, showing that the return levels estimated by TS are comparable to the ones obtained by these two methodologies. However, the TS approach comes with advantages on both SS and EA. With respect to SS the TS methodology uses the whole time series for the fit of the extreme value distribution, guaranteeing a more accurate estimation of large return levels. With respect to EA it decouples the detection of the non stationarity of the series from the fit of the extreme value distribution, involving a simplification of both the steps of the analysis. In particular the fit of the distribution can be accomplished by means of a simple MLE with a few degrees of freedom, simple to implement and to control. The detection of the non stationarity can be performed by means of easy-to-implement and fast-to-run low-pass filters which do not require as an input any parametric function for the variability, making this methodology well suited for massive applications where the simultaneous evaluation several time series is required.

An implementation of the TS methodology has been developed in an open-source matlab toolbox (tsEva), which is available at https://bitbucket.org/menta78/tseva.





## Appendix A

*Duality between the established approach and the TS methodology*

In this appendix we show that if the extremes of a time series *y(t)* are fitted by a non-stationary GEV $G_Y(y,t)$, then there is

a family of transformations $f(y,t): y(t) \rightarrow x(t)$ such that $G_Y(y,t) = G_X[f^{-1}(x,t)]$, where $G_X(x)$ is a stationary GEV

fitting the extremes of a supposed stationary series *x(t)*.

To prove this we expand relationship $G_Y(y,t) = G_X[f^{-1}(x,t)]$ finding:

$$\left\{1 + \varepsilon_x \left[\frac{f(y,t) - \mu_x}{\sigma_x}\right]\right\}^{1/\varepsilon_x} = \left\{1 + \varepsilon_y(t)\left[\frac{y - \mu_y(t)}{\sigma_y(t)}\right]\right\}^{1/\varepsilon_y(t)}, \tag{44}$$

where $[\varepsilon_y(t), \sigma_y(t), \mu_y(t)]$ are the time varying GEV parameters of $G_Y(y,t)$ and $[\varepsilon_x, \sigma_x, \mu_x]$ are the constant GEV

parameters of $G_X(x)$. Solving for $f(y,t)$ we find

$$f(y,t) = \frac{1}{\varepsilon_x}\left\{\sigma_x\left[1 + \varepsilon_y(t)\left(\frac{y - \mu_y(t)}{\sigma_y(t)}\right)\right]^{\frac{\varepsilon_x}{\varepsilon_y(t)}} - \sigma_x + \varepsilon_x\mu_x\right\}. \tag{45}$$

Equation (45) defines a family of functions because the values of the stationary GEV parameters $[\varepsilon_x, \sigma_x, \mu_x]$ can be

assigned arbitrarily. Furthermore if we chose $\varepsilon_x \neq 0$ then $f(y,t)$ is monotone in *y* for every time *t* and can therefore be

inverted, while for $\varepsilon_x = 0$ a Gumbel-specialized formulation can be derived from (44).

In the particular case of $\varepsilon_y = const. = \varepsilon_x$ function $f(y,t)$ reduces to

$$f(y,t) = \frac{y - \mu_y(t) + \mu_x/\sigma_x \cdot \sigma_y(t)}{\sigma_y(t)/\sigma_x}, \tag{46}$$

which is equivalent to formula (1) provided that $tr_y = \mu_y - \mu_x/\sigma_x \cdot \sigma_y$ and $ca_y = \sigma_y/\sigma_x$. Hence we can say that Eq. (1)

allows a general TS formulation for models with constant shape parameter, because we can arbitrarily impose $\varepsilon_x = \varepsilon_y$ in

(45) if we assume a constant $\varepsilon_y$. This finding is remarkable because it proves that any non stationary GEV model with

constant $\varepsilon_y$ can be connected to Eq. (1).

Equation (45) alone is not enough to formulate a fully generalized TS approach, because in (45) the non-stationary GEV

parameters $[\varepsilon_y(t), \sigma_y(t), \mu_y(t)]$ are regarded as known variables, which is a wrong assumption in practical applications.

But it is enough to say that any implementation of the non-stationary established approach is equivalent to a transformation

into a supposed stationary series *x(t)*. Therefore Eq. (45) could be used as a diagnostic tool for implementations of the

established approach: a condition for the validity of the non stationary model is that the transformed *x(t)* series is stationary.



## Acknowledgments

The authors would like to thank Simone Russo of the JRC and Francesco Fedele of the GIT for the precious suggestions. This work was co-funded by the JRC exploratory research project Coastalrisk and by the European Union Seventh Framework Programme FP7/2007-2013 under grant agreement no 603864 (HELIX: "High-End cLimate Impacts and eXtremes"; www.helixclimate.eu).

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





**Yearly maxima: trend only analysis**

| Error component (average) | West Ireland error (m) | Cape Horn error (m) | % ($err^2$) |
|---|---|---|---|
| **non stationary GEV** | | | |
| $std_y \cdot err(\sigma_x)$ | 0.0371 | 0.0372 | 100% |
| $err(std_y) \cdot \sigma_x$ | $5.876 \cdot 10^{-4}$ | $5.818 10^{-4}$ | <0.1% |
| $err(\sigma_y)$ | 0.0371 | 0.0372 | 100% |
| $std_y \cdot err(\mu_x)$ | 0.0538 | 0.0536 | 97.7% |
| $err(std_y) \cdot \mu_x$ | $3.6 \cdot 10^{-3}$ | $3.4 \cdot 10^{-3}$ | 0.4% |
| $err(tr_y)$ | $7.4 \cdot 10^{-3}$ | $7.0 \cdot 10^{-3}$ | 1.85% |
| $err(\mu_y)$ | 0.0538 | 0.054 | 100% |
| **non stationary GPD** | | | |
| $std_y \cdot err(\sigma_{GPDx})$ | 0.0418 | 0.0310 | 100% |
| $err(std_y) \cdot \sigma_{GPDx}$ | $1.12 \cdot 10^{-3}$ | $8.9 \cdot 10^{-4}$ | <0.1% |
| $err(\sigma_{GPDy})$ | 0.0418 | 0.0310 | 100% |
| $std_y \cdot err(u_x)$ | 0.1489 | 0.1376 | 100% |
| $err(std_y) \cdot u_x$ | $1.9 \cdot 10^{-3}$ | $1.7 \cdot 10^{-3}$ | <0.1% |
| $err(u_y)$ | 0.1491 | 0.1278 | 100% |

**Table 1: average error components for the non seasonal analysis of the GWWIII series for the locations of West Ireland and Cape Horn. The error is dominated by the component due to the stationary MLE.**



**Monthly maxima: seasonal analysis**

| Error component (average) | West Ireland error (m) | Cape Horn error (m) | % (err²) |
|---|---|---|---|
| **non stationary GEV** | | | |
| $std_{0y} \cdot sn_{std} \cdot err(\sigma_x)$ | 0.0135 | 0.0138 | 99.7% |
| $std_{0y} \cdot err(sn_{std}) \cdot \sigma_x$ | $7.2\cdot10^{-4}$ | $7.6\cdot10^{-4}$ | 0.3% |
| $err(\sigma_y)$ | 0.0135 | 0.0138 | 100% |
| $std_{0y} \cdot sn_{std} \cdot err(\mu_x)$ | 0.019 | 0.020 | 96.6% |
| $std_{0y} \cdot err(sn_{std}) \cdot \mu_x$ | 0.0014 | 0.0017 | 0.7% |
| $err(sn_{tr})$ | $4.86\cdot10^{-6}$ | $5.25\cdot10^{-6}$ | <0.1% |
| $err(\mu_y)$ | 0.0204 | 0.0214 | 100% |
| **non stationary GPD** | | | |
| $std_{0y} \cdot sn_{std} \cdot err(\sigma_{GPDx})$ | 0.025 | 0.029 | 100% |
| $std_{0y} \cdot err(sn_{std}) \cdot \sigma_{GPDx}$ | $9.4\cdot10^{-4}$ | $9.9\cdot10^{-4}$ | <0.1% |
| $err(\sigma_{GPDy})$ | 0.0253 | 0.0293 | 100% |
| $std_{0y} \cdot sn_{std} \cdot err(u_x)$ | 0.1061 | 0.1205 | 100% |
| $std_{0y} \cdot err(sn_{std}) \cdot u_x$ | 0.0011 | 0.0014 | <0.1% |
| $err(u_y)$ | 0.1063 | 0.1207 | 100% |

**Table 2: average error components for the seasonal analysis of the GWWIII series for the locations of West Ireland and Cape Horn. The error is dominated by the component due to the stationary MLE.**



| Return period | | 5 y | 10 y | 30 y | 100 y | 300 y |
|---|---|---|---|---|---|---|
| Rhine | NBI | -1.07%, | -1.51%, | -2.35%, | -3.43%, | -4.53% |
| (river dis.) | Max diff | -3.58%, | -4.40%, | -5.92%, | -7.81%, | -9.69% |
| Po | NBI | 1.47%, | 2.06%, | 2.92%, | 3.69%, | 4.25% |
| (river dis.) | Max diff | 5.87%, | 4.88%, | 5.60%, | 9.57%, | 13.06% |
| W. Ireland | NBI | -0.28%, | -0.14%, | 0.07%, | 0.27%, | 0.43% |
| (waves Hs) | Max diff | -0.91%, | -1.14%, | -1.48%, | 2.06%, | 2.51% |
| Cape Horn | NBI | -1.07%, | -1.13%, | -1.17%, | -1.18%, | -1.18% |
| (waves Hs) | Max diff | -1.87%, | -2.36%, | -3.12%, | -3.92%, | -4.59% |

**Table 3: normalized bias and maximum deviation between the return levels estimated with the TS methodology and the SS approach for the estimation of long term variations, for return periods of 5, 10, 30, 100 and 300 years, for the projected river discharge of the Rhine and the Po and for the projected significant wave height for West Ireland and Cape Horn.**



| Return period | | 5 y | 10 y | 30 y | 50 y | 100 y |
|---|---|---|---|---|---|---|
| La Spezia | NBI Winter | 1.19% | 1.51% | 1.95% | 2.14% | 2.39% |
| (waves Hs) | NBI Spring | 0.59% | 0.55% | 0.59% | 0.64% | 0.71% |
| | NBI Summer | 4.75% | 5.28% | 5.99% | 6.27% | 6.62% |
| | NBI Autumn | -1.17% | -1.03% | -0.78% | -0.66% | -0.50% |
| Ortona | NBI Winter | 3.74% | 4.23% | 4.91% | 5.20% | 5.57% |
| (waves Hs) | NBI Spring | 4.26% | 4.39% | 4.62% | 4.74% | 4.91% |
| | NBI Summer | -3.66% | -3.44% | -3.07% | -2.90% | -2.66% |
| | NBI Autumn | 1.41% | 1.45% | 1.59% | 1.68% | 1.81% |

**Table 4: normalized bias between the return levels estimated by the TS methodology and the EA methodology for the estimation of the seasonal variations, for return periods of 5, 10, 30, 50 and 100 years, for the four seasons, for the significant wave height in La Spezia and Ortona.**





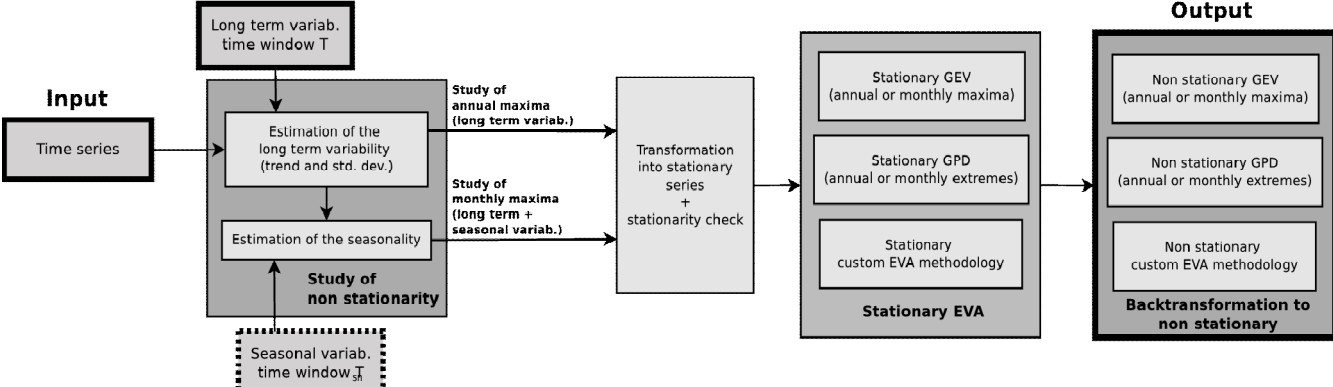

**Figure 1: TS methodology: block diagram.**




**Figure 2: non-seasonal analysis of the GWWIII projections of significant wave height for the location in West Ireland; (a): series, its trend and standard deviation; (b): the normalized series with higher order statistical indicators; (c): non-stationary GEV of annual maxima; (d): non-stationary GPD of annual peaks.**







**Figure 3: as in Figure 2 for the location near Cape Horn.**





**Figure 4: seasonal analysis of the GWWIII projections of significant wave height for West Ireland. Panel meaning as in Figure 2.**





**Figure 5: seasonal analysis of the GWWIII projections of significant wave height for Cape Horn. Panel meaning as in Figure 2.**







**Figure 6: non-seasonal analysis of the residual water levels modeled for the Hebrides islands. Panel meaning as in Figure 2.**



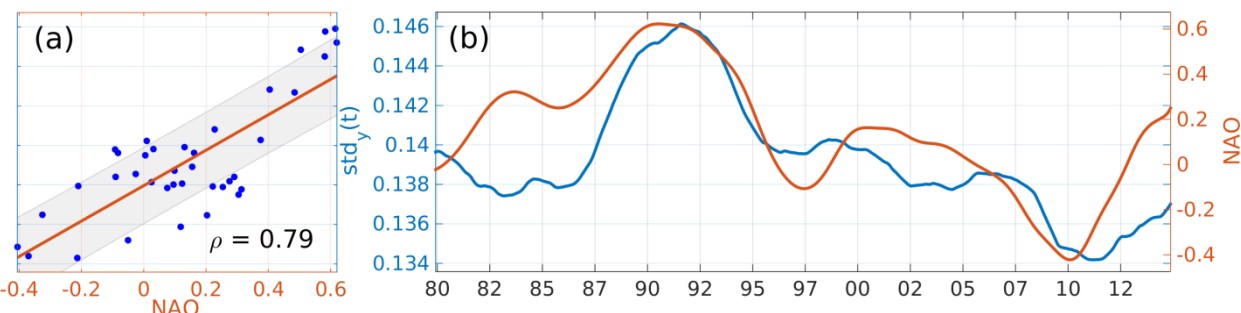

**Figure 7: time varying standard deviation std$_y$(t) estimated by means of the TS methodology versus the yearly average of the NAO index, scatter plot (a) and time series (b).**





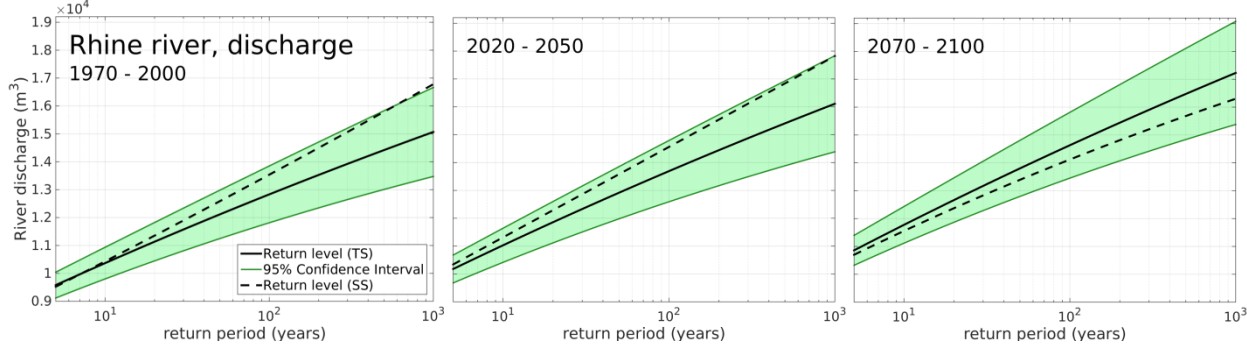

**Figure 8: return level plots for the discharge of the Rhine river at its mouth, TS methodology (black continuous line), 95% confidence interval for the TS methodology (green band) and SS (black dashed line), for the time slices 1970-2000, 2020-2050 and 2070-2100.**





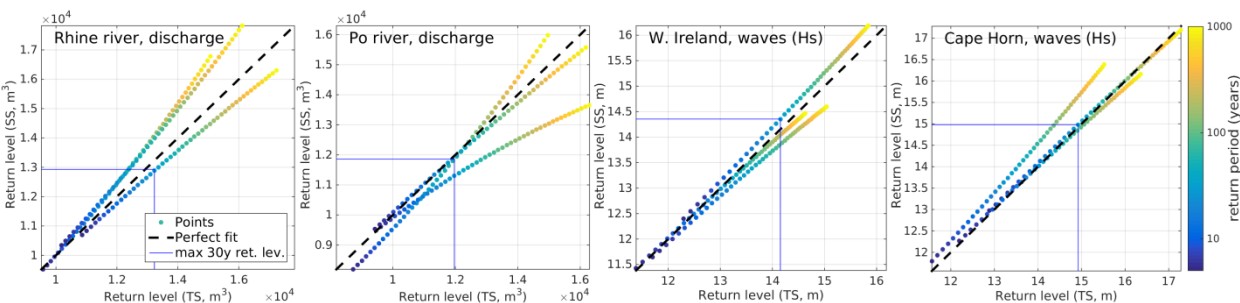

**Figure 9: return levels modeled by the TS methodology (x axis) vs those modeled by the SS methodology (y axis) for the discharge of the Rhine and Po rivers and the significant wave height in West Ireland and Cape Horn. The three series of dots represent the three time slices.The color of the dots represents the return period. The blue lines represent the maximum 30 years return level.**





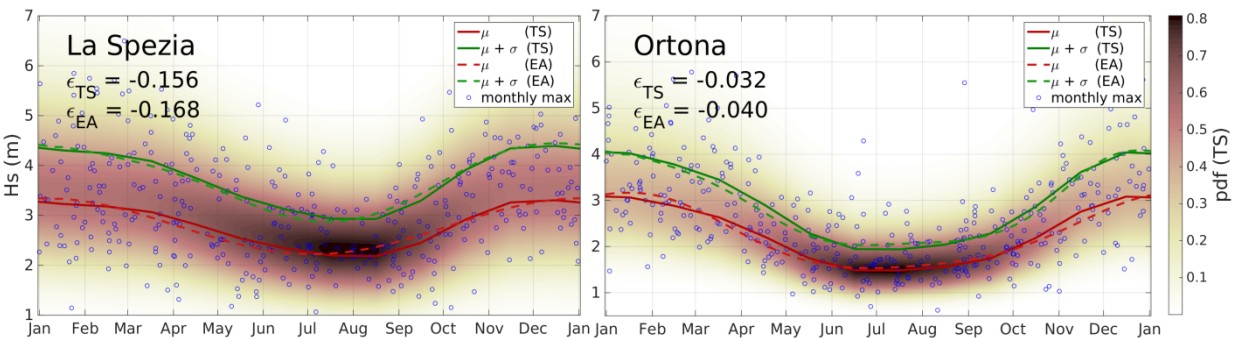

**Figure 10: seasonal cycle estimated by TS and by EA for the series of significant wave height of La Spezia and Ortona.**



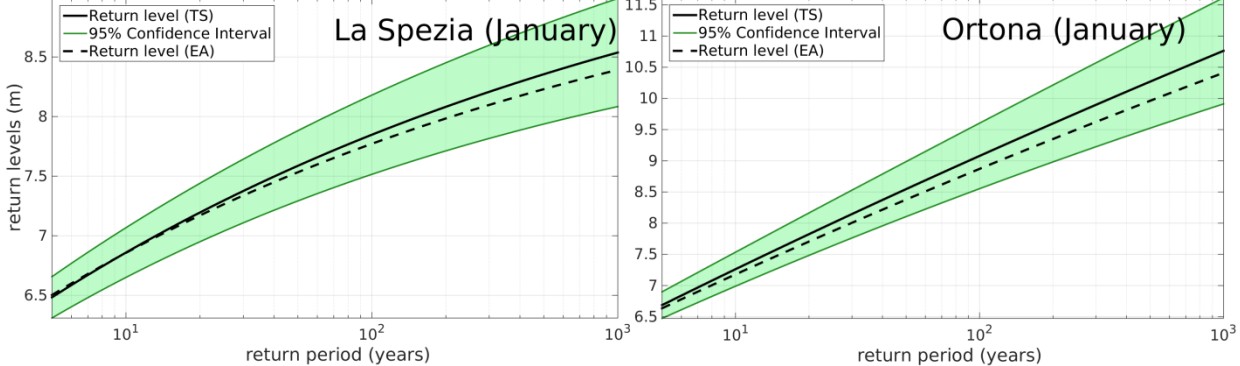

**Figure 11: return levels for La Spezia and Ortona for the month of January, estimated by the TS methodology (black continuous line) and by the established approach (black dashed line labeled as EA). The green area represents the 95% confidence interval estimated by the TS approach.**

