# Peer review of "The transformed-stationary approach: a generic and simplified methodology for non-stationary extreme value analysis"

_Hydrology and Earth System Sciences, 2016_

## Referee Comment (RC1) · Anonymous Referee #1 · 31 Mar 2016

The present paper provides a methodology to estimate extreme values from non stationary time series data. The methodology is well explained and documented and is adequately compared with other methods that normally used for non-stationary data. It has to be mentioned that the approach is mainly applicable to forecast or hintacast data because it is designed for very long time-series.

The paper is very well written and with good and extensive documentation of the statistical methodology. Furthermore, the method is applied to 3 time series of different geophysical data. I believe that the paper is interesting and of significant scientific quality and I am suggesting it for publication.

As a general comment I would say that the mathematical documentations is a bit extensive but in line with the presentation of a new mathematical method.

Some minor comments are presented below than can improve

Page 4 Line 6: MLE is already defined in page 2

Page6 line 23: What is : sn (t) probably you mean std (t)

Page 8 lines 15-25: If I am not mistaken the authors describe the methodology of calculating the seasonal anomalies, i.e. the deviations of the monthly data from a given climatology. If this is the case please state, on the contrary please indicate the differences and the error differences with e standard methodology. The inclusion of the equations is not necessary since an open source code is available but I agree that may help in the implementation.

Figure 1: In the season variability time window the 'sn' is misplaced

Page 12 line 24: Transformation 1. Do you mean Transformation using Eq (1)?

---

## Referee Comment (RC2) · Anonymous Referee #2 · 2 Apr 2016

This article introduces a transformed-stationary (TS) method for extreme value analysis in earth science. Authors did a good job illustrating the specific procedures of the TS approach. Tables and figures speak for themselves with titles and labels. The results comparison among three methods – TS approach, established approach (EA), and stationary on slice (SS) approach – demonstrated that TS method is sufficient for the estimates of distribution parameters and return levels when adopting EA as the benchmark.

The uncertainty in extreme value analysis can be very large even without non-stationarity. Besides using EA as a benchmark, it would be better if the uncertainties (bias or standard errors) of the estimator (for either the distribution parameters or return levels) are also compared among three approaches.

The use of English language is not perfect. Some sentences are too long and hard to understand. The use of prepositions in some phrases/sentences are awkward.

Specific comments in the text:

1. Page 5 line 1. Add "as" in the middle of the sentence. "It can be shown that the time-dependent GEV parameters given by Eq.(7) are the same" as "that would be obtained from…"

2. Page 6 line 1. The notation x in mu_x should be a subscript.

3. Page 5 line 3. "it is maximum also" –> it is also the maximum

4. Page 9 footnote line 1. Suspected grammar error in the first half sentence.

5. Page 12 line 3-4 citation format. "implemented by (Alfieri et al., 2015) and (Vousdoukas et al., 2016)" –> implemented by Alfieri et al. (2015) and Vousdoukas et al. (2016)

6. Page 14 line 11-12. "The estimated seasonal GEV and GPD are significantly lower than…" Does the "estimated GEV/GPD" refer to estimated pdf or estimated return of levels? The text is not clear enough.

7. Page 19 conclusion. The generality of TS method has been described in the first paragraph in session 5 (page 17). It seems redundant.

8. Figure 1. Resolution is not high enough (based on the size of 100% in PDF file).

---

## Author Comment (AC1) · 15 Apr 2016

First of all we would like thank the editor for his interest in our work, and reviewer 1 for his time spent reading carefully the paper, and for the comments and the suggestions to improve the quality of the work. Follows a item-by-item reply to the reviewer comments.

reviewer: Page 4 Line 6: MLE is already defined in page 2

author: The second definition has been removed

reviewer: Page6 line 23: What is : sn (t) probably you mean std (t)

author: Corrected as indicated by the reviewer.

reviewer: Page 8 lines 15-25: If I am not mistaken the authors describe the methodology of calculating the seasonal anomalies, i.e. the deviations of the monthly data from

a given climatology. If this is the case please state, on the contrary please indicate the differences and the error differences with e standard methodology. The inclusion of the equations is not necessary since an open source code is available but I agree that may help in the implementation.

author: This is what we mean with formula 21. Thanks to the reviewer for suggesting a clearer explanation of the meaning of the formula.

reviewer: Figure 1: In the season variability time window the 'sn' is misplaced.

author: Corrected as indicated by the reviewer.

reviewer: Page 12 line 24: Transformation 1. Do you mean Transformation using Eq (1)?

author: Yes. Changed with "transformation given by Eq(1)".
* * *

---

## Author Comment (AC2) · 15 Apr 2016

First of all we would like thank the editor for his interest in our work, and reviewer 2 for his time spent reading carefully the paper, and for the comments and the suggestions to improve the quality of the work. Follows a item-by-item reply to the reviewer comments.

reviewer: The uncertainty in extreme value analysis can be very large even without nonstationarity. Besides using EA as a benchmark, it would be better if the uncertainties (bias or standard errors) of the estimator (for either the distribution parameters or return levels) are also compared among three approaches.

author: Thank you for the suggestion. The mean confidence intervals for the return levels estimated by TS, SS and EA have been added in tables 3 and 4, and have been commented in paragraphs 4.1 and 4.2.

reviewer: 1. Page 5 line 1. Add "as" in the middle of the sentence. "It can be shown that the timedependent GEV parameters given by Eq.(7) are the same" as "that would be obtained from. . ."

author: Amended as requested

reviewer: 2. Page 6 line 1. The notation x in mu_x should be a subscript.

author: Corrected

reviewer: 3. Page 5 line 3. "it is maximum also" –> it is also the maximum

author: Amended as requested

reviewer: 4. Page 9 footnote line 1. Suspected grammar error in the first half sentence.

author: The sentence was reformulated hoping to make it clearer: We can evaluate the error on the average of the observations by propagating the intrinsic error of each observation, which is given by the standard deviation of s, to expression ...

reviewer: 5. Page 12 line 3-4 citation format. "implemented by (Alfieri et al., 2015) and (Vousdoukas et al., 2016)" –> implemented by Alfieri et al. (2015) and Vousdoukas et al. (2016)

author: Amended as requested

reviewer: 6. Page 14 line 11-12. "The estimated seasonal GEV and GPD are significantly lower than. . ." Does the "estimated GEV/GPD" refer to estimated pdf or estimated return of levels? The text is not clear enough.

author: In the specific we refer to the pdf. Clarified as requested

reviewer: 7. Page 19 conclusion. The generality of TS method has been described in the first paragraph in session 5 (page 17). It seems redundant.

author: The reviewer is right that this concept is repeated. However we prefer to stress it once again in the conclusions, as we regard this as a major aspect of the TS approach.

reviewer: 8. Figure 1. Resolution is not high enough (based on the size of 100% in PDF file).

author: Figure 1 has been replaced and the resolution increased.

---

## Author Comment (AC4) · 15 Apr 2016

Dear editor and reviewers, for the sake of completeness I attached to this message a "final" version of the reviewed paper, which is identical to the revision version, but all the revisions have been accepted.

Please also note the supplement to this comment:
http://www.hydrol-earth-syst-sci-discuss.net/hess-2016-65/hess-2016-65-AC4-supplement.pdf

---

## Author Response (AR1)

First of all we would like thank the editor for his interest in our work, and the reviewers for their time spent reading carefully the paper, and for the comments and the suggestions to improve the quality of the work. Follows a item-by-item reply to the reviewer comments.

**Reviewer 1:**
The present paper provides a methodology to estimate extreme values from non stationary time series data. The methodology is well explained and documented and is adequately compared with other methods that normally used for non-stationary data. It has to be mentioned that the approach is mainly applicable to forecast or hintacast data because it is designed for very long time-series.
The paper is very well written and with good and extensive documentation of the statistical methodology. Furthermore, the method is applied to 3 time series of different geophysical data. I believe that the paper is interesting and of significant scientific quality and I am suggesting it for publication.
As a general comment I would say that the mathematical documentations is a bit extensive but in line with the presentation of a new mathematical method.
Some minor comments are presented below than can improve

Page 4 Line 6: MLE is already defined in page 2
  *The second definition has been removed*

Page6 line 23: What is : sn (t) probably you mean std (t)
  *Corrected as indicated by the reviewer.*

Page 8 lines 15-25: If I am not mistaken the authors describe the methodology of calculating the seasonal anomalies, i.e. the deviations of the monthly data from a given climatology. If this is the case please state, on the contrary please indicate the differences and the error differences with e standard methodology. The inclusion of the equations is not necessary since an open source code is available but I agree that may help in the implementation.
  *This is what we mean with formula 21. Thanks to the reviewer for suggesting a clearer*
  *explanation of the meaning of the formula.*

Figure 1: In the season variability time window the 'sn' is misplaced.
  *Corrected as indicated by the reviewer*.

Page 12 line 24: Transformation 1. Do you mean Transformation using Eq (1)?
  *Yes. Changed with "transformation given by Eq(1)".*

**Reviewer 2:**
This article introduces a transformed-stationary (TS) method for extreme value analysis in earth science. Authors did a good job illustrating the specific procedures of the TS approach. Tables and figures speak

for themselves with titles and labels. The results comparison among three methods – TS approach, established approach (EA), and stationary on slice (SS) approach – demonstrated that TS method is sufficient for the estimates of distribution parameters and return levels when adopting EA as the benchmark.

The uncertainty in extreme value analysis can be very large even without nonstationarity. Besides using EA as a benchmark, it would be better if the uncertainties (bias or standard errors) of the estimator (for either the distribution parameters or return levels) are also compared among three approaches.

> *Thank you for the suggestion. The mean confidence intervals for the return levels estimated by TS, SS and EA have been added in tables 3 and 4, and have been commented in paragraphs 4.1 and 4.2.*

The use of English language is not perfect. Some sentences are too long and hard to understand. The use of prepositions in some phrases/sentences are awkward.
Specific comments in the text:

1. Page 5 line 1. Add "as" in the middle of the sentence. "It can be shown that the timedependent GEV parameters given by Eq.(7) are the same" as "that would be obtained from. . ."

> *Amended as requested*

2. Page 6 line 1. The notation x in mu_x should be a subscript.

> *Corrected*

3. Page 5 line 3. "it is maximum also" –> it is also the maximum

> *Amended as requested*

4. Page 9 footnote line 1. Suspected grammar error in the first half sentence.

> *The sentence was reformulated hoping to make it clearer:*
> *"We can evaluate the error on the average of the observations by propagating the intrinsic error of each observation, which is given by the standard deviation of s, to expression ..."*

5. Page 12 line 3-4 citation format. "implemented by (Alfieri et al., 2015) and (Vousdoukas et al., 2016)" –> implemented by Alfieri et al. (2015) and Vousdoukas et al. (2016)

> *Amended as requested*

6. Page 14 line 11-12. "The estimated seasonal GEV and GPD are significantly lower than. . ." Does the "estimated GEV/GPD" refer to estimated pdf or estimated return of levels? The text is not clear enough.

> *In the specific we refer to the pdf. Clarified as requested*

7. Page 19 conclusion. The generality of TS method has been described in the first paragraph in session 5 (page 17). It seems redundant.

*The reviewer is right that this concept is repeated. However we prefer to stress it once again in the conclusions, as we regard this as a major aspect of the TS approach.*

8. Figure 1. Resolution is not high enough (based on the size of 100% in PDF file).

*Figure 1 has been replaced and the resolution increased.*

[revised manuscript text omitted]

---

## Author Response (AR2)

Dear Editor,

First of all we would like to thank you and the reviewers for your work and the observations to improve the quality of the paper and to make it more conformal to the required standard. We put efforts to meet all the guidelines as requested. In particular:

- References to sections have been abbreviated as needed.
- References to equations have been abbreviated as needed.
- The footnote at page 9 has been removed, and its content inserted into the text.
- Figure and table captions have been made stand-alone, all the content of the figures have been described, the abbreviations have been explained the first time in each caption, but for the very common ones (like GEV and GPD).
- The whole manuscript has been carefully reviewed by an English mother language.

We hope now the manuscripts meets all the required standards.

Best regards,

Lorenzo

---

## Author Response (AR3)

Dear Editor,

Thank you again for your accurate contribution to improve this work. A further effort has been done to meet all the guidelines as requested, and to correct the mistakes you listed and to find other mistakes.

If still possible we would like to change the title of the manuscript into "The transformed-stationary approach: a generic and simplified methodology for non-stationary extreme value analysis". We think the new title would better capture the content and open the article (hence also the journal) to a wider audience.

Follows a item by item reply to your comments:

1) *Editor: I previously asked the authors to please review the HESS manuscript guidelines to ensure that the manuscript was in compliance; however, there are many places where the guidelines have not been followed - particularly for equations and symbols. Specific guidelines for equations and symbols are below. The authors must take note particularly for items (a), (b), and (d). The notation used in the equations and text need to be revisited based on the guidelines.*

   Authors: The notation of the equations has been completely revised. In particular, (a) the numeration of the equation has been revised and long equation have been splitted. (b) Multiletter functions such as tr or std have been replaced with single letter ones such as T and S. A few remaining multiletter functions, such as "month", in eq. 32, or Err in the section about the statistical error have been set to roman. (d) Single letter variables/functions have been set to italic everywhere, while the subscripts have been everywhere styled to roman.

2) *Please review again the requirements for figures. See last sentence below in the guidelines for figure composition.*
   *"Figure composition: It is important for the production process that separate figures are submitted. Composite figures containing multiple panels should be collected into one file before submission. The figures should be labelled correctly with Arabic numerals (e.g. fig01, fig02). They can be submitted in \*.pdf, \*.ps, \*.eps, \*.jpg, \*.png, or \*.tif format and should have a resolution of 300 dpi. The width should not be less than 8 cm. A legend should clarify all symbols used and should appear in the figure itself, rather than verbal explanations in the captions (e.g. "dashed line" or "open green circles")."*

   Authors: Separate files for the images have been prepared and accordingly uploaded to the system. We apologize, we did not have the possibility to upload them before now. A legend has been added to the scatter plot of figure 7, which was the only figure without a comprehensive legend.

3) *The grammar still needs improving. It is disappointing that there are so many errors remaining after several revisions. I recommend the authors read the manuscript again to improve clarity. Here are only some of the changes that are needed based on my reading. Please note that is only a partial list. Submit a version of the revised manuscript showing track changes so that I can confirm these changes were made and see evidence that other areas of the text were reviewed and revised by the authors.*

We thank the editor for her suggestions and contribution to improve the clarity and correctness of the manuscript. All of her suggestions have been accepted. Follows a item-to-item rebuttal for each of the proposed changes.

*p. 1,*

*l. 21: 'ones' to 'results'*
Amended as requested
*l. 23 and 25: 'it' to 'the proposed technique'*
Amended as requested
*l. 25 'values' to 'value'*
Amended as requested

*p. 2*

*l. 12: add so that sentence reads "latitudes on a global scale"*
Amended as requested
*l. 23: add so that sentence reads "referred to as the established approach"*
Amended as requested
*l. 32: delete 'approach'*
Amended as requested

*p. 3*

*l. 5: sentence should read, "This technique is referred to in the text as…"*
Amended as requested
*l. 21: You have already spelled out EVA earlier so only use abbreviation here.*
Amended as requested

*p. 4*

*l 4: should read "has mean equal to zero and variance equal to 1"; the use of 'null average' is awkward*
Amended as requested
*l 9: Move 'for example' in front of "A simple test"*

Amended as requested

*l 11: The use of GEV and Gx is confusing. Choose one notation for the GEV and use throughout the manuscript*

As suggested, we substituted Gx with GEVx , Gx with GEVy everywhere in the text.

*l 15: an example of where x and y are not italics and should be according to the submission guidelines*

We put effort into finding all the reference to variables in the text and italicized them.

*p. 5*

*l 8: Pgns does not appear to be defined*

Thank you for suggesting us to clarify this point. The meaning of the Pgns has been clarified at lines 5 and 10 of page 5.

*p. 6*

*l 5: x should be italics, add comma at the end of pgx(x), define "it", change "that" to "thusly"*

Substituted "This means that" with "Therefore". The rest have been amended as requeted

*l 8: EA and EVA are very close to each other in abbreviation and could lead to confusion by readers. Consider changing. Remove quotes around 'dual'*

Done. Substituted everywhere EA with Established Method (EM)

*l. 17 x and y are not italics*

Amended

*p. 7*

*l 3: according to the manuscript guidelines, Eq. should be placed in front of (2)*

Amended

*l. 8: put commas around "respectively"*

Done

*Question about notation: What does the '0' in the subscripts mean?*

In the notation the subscript "0" denotes the long-term varying components. Added the explanation to the manuscript.

*p. 8*

*l 4: "about" to "of"*

Amended as requested

*l 5: delete "the" in "variations in the climate"*

Amended as requested

*l. 6: check guidelines if a posteriori needs to the italics. Also, the dash should be removed.*

Amended as requested

*p. 9*

*l 5: "rough" is capitalized in the equations and should be also capitalized in the text*
Done
*l. 8: Commas around "in general"*
Amended as requested

*p. 12*

*l 15: Annual maxima of what?*
Of a considered time series. Explained as requested
*l. 28: Should read "GEV and GPD"*
Amended as requested
*l 28: Should read "Eqs. (7) and (15), respectively"*
Amended as requested

*p. 13, l 13: What figure are you referring when you point to panels (c) and (d)?*
Of Figure 2 and Figure 3. Explained in the text

*p. 14*

*l 2: Should read: "error associated with the stationary…"*
Amended as requested. Expression "associated to" has been corrected everywhere.

*l 6: delete comma*
Amended as requested